# The Surface Integrity of Titanium Alloy When Using Micro-Textured Ball-End Milling Cutters

**DOI:** 10.3390/mi10010021

**Published:** 2018-12-29

**Authors:** Shucai Yang, Song Yu, Chunsheng He

**Affiliations:** College of Mechanical and Power Engineering, Harbin University of Science and Technology, Harbin 150080, China; yangshucai@hrbust.edu.cn (S.Y.); 0gattuso0@gmail.com (C.H.)

**Keywords:** micro-texture, ball-end milling cutter, titanium alloy, surface roughness, work hardening

## Abstract

Processing certain kinds of micro-textures onto the surface of tools can improve their wear resistance, reduce the friction between them and machined surfaces, prolong their service life and improve their processing efficiency. When milling titanium alloy with ball-end milling cutters, the cutting force and the cutting heat causes plastic deformation and a concentration of stress on workpiece surfaces, damaging their surface integrity. In this paper, we report on a test involving the milling of titanium alloy, where a micro-texture was placed onto the front of a ball-end cutter and the surface roughness and work hardening of the machined surface were studied. The orthogonal experiment was designed around changes in the diameter of the micro-texture, its depth, the spacing between individual micro-pits, and its distance from the cutting edge. Data from the experiment was then used to assess the influence changes in the micro-texture parameters had upon the roughness and hardening of the surface. The data was processed and analyzed by using regression analysis and a prediction model for surface roughness and work hardening was established. The reliability of the model was then verified. The contents of this paper provide a theoretical basis for improving the cutting performance and the surface machining quality of cemented carbide tools.

## 1. Introduction

Titanium alloy has become one of the main structural materials used in the aerospace industry because of its outstanding characteristics, such as a high specific strength, and good corrosion resistance. The proportion of titanium alloy that is used in advanced aerospace engines is increasing rapidly. However, the friction coefficient between titanium alloy and the tools that cut it is large, and the friction speed for titanium alloy chips is particularly high along the rake face [1]. Severe friction causes tool wear, and results in poor surface quality. In the face of this, a large number of studies have found that adding a micro-texture to the tool’s surface can bring about good anti-friction and anti-adhesion effects, improve wear resistance, and improve the quality of the machined surface. This discovery provides a new research direction for the development of cutting tools [2].

Based on the theory of bionic tribology, the surface micro-textured cutting tool is a new type of tool which can improve the cutting state of the tool by machining a certain regular shape onto the tool’s rake face. Many scholars have fabricated many kinds of surface textures from micron to nanometer on cemented carbide and high-speed steel cutting tools by electrical discharge machining (EDM) [3], photolithography [4], and laser processing technology [5], and have studied the cutting performance. It is concluded that the micro-pit texture has the best wear-reducing and wear-resisting properties, and that the cutting performance of the cutting tools is improved the most [6].

An assessment of whether titanium alloy parts are of sufficient quality not only relates to machining accuracy, but also the machined surface integrity. Surface integrity refers to the status of the surface layer after mechanical processing [7]. This includes surface roughness, surface hardening, surface residual stress, and the surface metamorphic layer. At present, it is attracting a great deal of research, and a number of studies have attempted to establish surface integrity analysis and evaluation models and to explore the surface formation process. However, research on the effect of the cutting tools on the surface integrity after processing is relatively small.

Bordin et al. [8] have studied the regular patterns of influence between the cutting parameters and the surface integrity-related parameters (surface roughness, residual stress, and work hardening) when turning CoCrMo alloy under dry-cutting conditions. With regard to establishing prediction models of surface roughness, Chen et al. [9] have analyzed the influence of a tool’s cutting edge shape in relation to the surface residual height formation, and have developed techniques for predicting surface roughness that are based on neural networks. Wang [10] used single-point diamond cutters to carry out the ultra-precise turning of aluminum alloy, and based on a multiple regression analysis method, developed a prediction model of surface roughness that can identify the regular effects of cutting parameters on surface roughness. By using a response surface method, Wu [11] has established a prediction model with multiple regression for surface roughness, and has verified the accuracy of the model by using variance analysis.

Looking more specifically at surface hardness, Che-Haron and Jawaid [12] found that the hardened case depth of a Ti6Al4V turned component was as high as 100 microns, higher than that of the material itself. The micro-hardness was about 200 microns, however, which is lower than the hardness of the material. Canteroa [13] has pointed out that the hardness of the deformation layer near a drilled surface shows an increase of about 30%. Chen and Hu [14] have studied the depth of the hardened layer for titanium alloy Ti10V2Fe3Al. Their results show that, when milling titanium alloy with cemented carbide tools, newer tools are not as hard as worn tools and climb milling produces a shallower hardened layer than inverse milling. Zhou [15] tried using coated carbide cutters for the end-milling of titanium alloy TB6 and found that the degree of work hardening was significantly reduced when the milling speed was increased, with feed volume and depth also having a significant effect. However, as tools become worn, they produce a softening phenomenon instead, which becomes more apparent as the degree of wear increases.

Arizmendi [16] presented a model for the topography prediction of ball-end milled surfaces, considering the tool parallel axis offset. First, the equations of cutting edges trajectories and the envelope equation of the material swept by the tool are derived. Later, the trajectories are cut by planes that are perpendicular to the feed direction, obtaining a set of transcendental equations that are solved by transforming them to polynomial equations through Chebyshev expansions. This procedure presents an advantage over previous models in the literature, by not requiring any starting point to achieve the solution. Finally, experimental results are presented and compared to the model predictions. Lacalle [17] applied ball burnishing onto sculptured surfaces, aiming at enhancing surface roughness. Different strategies are possible for burnishing, with continuous burnishing (CB), which uses a five-axis interpolation of the machine tool, and patch burnishing (PB), which uses a simpler 3 + 2 axis interpolation. Using both techniques, complex parts are burnished, and a big improvement in surface roughness is achieved, but some differences between both approaches appear. Two parts have been previously machined in a five-axis milling center, and finished by using the ball burnishing approaches. The first one is a steel AISI 1045 with a hemisphere shape, whose geometry is simple. The second one is a steel DIN 1.2379 part (64 HRC), with more complex features. The surface quality was evaluated for both burnishing approaches, obtaining significant improvements on the surface roughness and hardness. The main general conclusion is that ball burnishing reduces the roughness without penalizing the manufacturing time or surface integrity, and therefore, it is suitable for complex surfaces. Artetxe [18] described the implementation of a cutting force prediction model that introduces the radial engagement reduction caused by (i) tool runout and (ii) workpiece flexibility. The model is used as the seed for a calculation module, based on solid modeling, to obtain the surface roughness and topography by applying the cutting force-wall restitution equilibrium, making use of the power of computer aided manufacturing (CAM) solid modelers for the final topography definition. This idea takes advantage of the good accuracy of current solid modelers in comparison with the z-map topographical approach for obtaining the cutter-workpiece engagement (CWE) at each toolpath step. Several flank milling tests were carried out to validate the simulations. The results demonstrated a significant level of accuracy to predict shape errors. The model may be integrated in a CAM procedure and it is a useful utility for reducing deformation on the manufacturing of blisks and impellers. Huissoon [19] investigated the effects of various controllable polishing parameters on the resulting surface when using a flexible abrasive disk on die steel. The objective is to achieve a robust process that results in a consistent surface finish, the roughness of which can be specified and controlled. Experimental results indicate that the inclination angle of the disk with respect to the workpiece, and the feedrate have optimal values that minimize the variability of the surface finish, while the normal force applied to the disk should be used to control the nominal surface roughness

In this paper, we report on a study of the surface integrity of titanium alloy after it has been cut by a micro-textured ball-end milling cutter. An orthogonal milling test was carried out that focused upon the impact of changing various micro-texture parameters. The relationship between the micro-texture parameters and surface roughness and work hardening was analyzed, providing insight into the mechanisms by which these phenomena are produced.

## 2. The Factors Affecting Titanium Alloy Surface Quality When Milled by a Micro-Textured Ball-End Milling Cutter

### 2.1. The Machined Surface Formation Process

In the process of cutting, the cutting edge should not be too sharp. Instead, there should be a small blunt circle radius (*r_n_*). In addition, over the course of cutting, the tool face becomes worn, forming a narrow edged surface with a wear width of VB. This makes the third deformation zone more complicated [20]. In view of these issues, when analyzing the formation of the machined surface, it is necessary to take into account the blunt circle radius and the effect of wear on the rear face [21]. The cutting effect of a ball-pointed milling cutter’s single point cutting edge resembles oblique cutting. Oblique cutting can itself be simplified to right-angle cutting. In this paper, our analysis is based on a right-angle cutting model, as shown in Figure 1.

Referring to Figure 1, as a result of the blunt circle radius rn on the cutting edge, when the cutting layer passes through point O, the cutting depth *a_c_* cannot be completely removed by the cutting tool. The thickness of the residual part here is *a_c_*. This is because when the metal for cutting approaches the blade in direction *v_c_*, extrusion and shear deformation occur, and a chip is formed as it slips in OM direction across the shear surface [22]. The plastic deformation of the part of the material below O is a result of the extrusion friction produced by the blunt circle radius of the cutting edge. At the same time, elastic deformation occurs inside the matrix. When the surface of the tool is no longer in contact with the surface of the workpiece, the elasticity of the workpiece is restored to *h*. During this process, the contact length between the workpiece and the rake face changes from VB to VB + CD, and the extrusion and friction between the flank and the workpiece surface is increased. This results in the increased deformation of the machined surface and can produce hardening.

### 2.2. The Influence of a Micro-Textured Ball-End Milling Cutter on the Surface Quality of the Workpiece

#### 2.2.1. Factors Affecting the Surface Quality

In the cutting processes, the contact between the tool and the workpiece and the cutter and the chip leads to friction and deformation, together with shear slip in the deformation zone. The small contact area at the edge of a cutting tool is subjected to great amounts of pressure, high temperature, and intense friction. This part of the cutter therefore experiences diffusion wear, adhesion wear, oxidation wear, and abrasive wear, all of which lead to changes in the cutting tool’s shape and size, thus affecting the surface quality of the parts being cut. So, to ensure that tools are effectively designed, manufactured to a high enough quality and correct clamped, it is important to choose the right geometric factors. To handle the surface cutting conditions, and to maximize tool life, we need to improve the production efficiency and to reduce the processing costs. At the same time, whether the choice of the cutting amount is reasonable makes a great deal of difference to the surface integrity of the parts. A correct choice of cutting parameters maintains the full use of both the cutting performance of a machine tool, and its dynamic performance (power and torque), and obtains a higher productivity and lower cutting cost, whilst preserving the integrity of the surface.

#### 2.2.2. Effect of the Micro-Textured Ball-End Milling Mechanism on the Quality of Machined Surfaces

The surface quality consists of two parts. One is the micro-surface morphology, such as the surface roughness or waviness. The other is the physical and chemical properties of the surface, such as residual stress or corrosion resistance, etc. When a ball-end milling cutter is used for milling titanium alloy, the cutting angle of the tool is continuously changing, because of the spherical shape of the cutter head. As a result, the milling force is also continuously changing, though only within a certain range, and without much mutation. This guarantees the surface quality of the workpiece. However, in the process of finishing, the continuously changing milling force can cause the plastic deformation of the machined surface, resulting in larger degrees of surface roughness and a lower workpiece surface quality.

When a micro-texture is placed on a tool’s surface, the friction between the rake surface and the surface to be machined is weakened as a result of the micro-pits. At the same time, micro-pits can play a role in the secondary breaking-up of the chips and the storage of coolant, thus helping to keep a machined surface sufficiently cool. In this way, the impact of the continuously changing milling force on the roughness of the machined surface can be reduced, and the surface quality of the workpiece is improved.

## 3. Methods for Testing Work Hardening and Evaluation Criteria

### 3.1. Evaluation Criteria for the Work Hardening

In the course of processing the surface of a metal workpiece, the cutting force can produce strong plastic deformation, metal lattice distortion, elongation of grains, fibrosis, and broken strands, resulting in an increase in the strength and hardness of the metal’s surface, and a reduction in its plasticity. This phenomenon is called workpiece surface cold metal hardening [23]. On the other hand, the heat generated by the cutting process increases the temperature of the workpiece surface. When the temperature rises to a critical point, the strengthened metal returns to its normal state, and the physical and mechanical properties of the hardening are eliminated. This is referred to as a softening phenomenon. In that case, work hardening depends on the relative rates for the hardening speed and the softening speed [24]. Indexes for evaluating work hardening include the surface microhardness (*HV*) of the layer, the depth of the hardened layer *h* and the degree of hardening *N*. They can be expressed together as follows:(1)$$N=\frac{HV-HV_{0}}{HV_{0}}\times 100\%$$
where *HV*_0_ is the original microhardness of the metal.

### 3.2. Approaches to Testing for Work Hardening

The test method for work hardening [25] is that, under a prescribed test force, a four-faced pyramidal positive diamond with two opposite angles at the bottom of 136° has to be pressed into the surface of the test material, and held there for a prescribed time. After the test force has been removed the diagonal length of the indentation on the surface of the specimen is measured. The corresponding microhardness value can be calculated using the hardness formula below (2) [26]. The indentation method is illustrated in Figure 2.
(2)$$HV=0.102\frac{2F\sin\;\frac{136^{{}^{\circ}}}{2}}{d^{2}}\approx 0.1891\frac{F}{d^{2}}$$
where, *F* is the load applied (*N*); *d* is the indentation’s diagonal length (μm); and *HV* is the surface microhardness (Vivtorinox hardness).

The method for testing microhardness involves taking a cross-section because the size of the actual workpiece is large. For the purposes of the test the workpiece has to be cut perpendicular to its surface, and parallel to the feed direction. In our test the specimen was cut into 10 mm × 10 mm × 7 mm blocks. An HXD-1000 microhardness tester (Shanghai Taiming Optical Instrument Co. Ltd., Shanghai, China) was used to detect the microhardness in the segments after cutting. We were then able to analyze the influence of the parameters relating to different kinds of micro-pit-based texture on the machine-hardened surface. Figure 3 shows the titanium alloy samples used in the experiment.

## 4. High Speed Milling of Titanium Alloy with A Micro-Textured Ball-End Milling Cutter

### 4.1. Cutting Tools and Workpiece Materials

The workpiece material used for the test was α + β Ti6Al4V with a size of 150 mm × 10 mm × 80 mm. The workpiece was clamped to a specific fixture with a 15° slope. The physical and chemical components of α + β Ti6Al4V are shown in Table 1 and Table 2. The tool used in the test was a Dijet BNM-20 ball-end milling cutter (DIJET INDUSTRIAL CO., LTD, Osaka, Japan), arbor model reference BNML-200105S-S20C. It is made entirely of hard alloy, and it has a tool grade of YG8. The blade and the arbor are shown in Figure 4.

### 4.2. Test Equipment and Test Parameters

The equipment used in the experiment was a VDL-1000E type 3-axis NC milling machine (Dalian machine tools group, Dalian, China). The milling modes were down-milling, unidirectional and line milling. The processing parameters were as follows: cutting speed *v_c_* = 120 m/min; feed per tooth *f_z_* = 0.08 mm/z; cutting thickness *a_p_* = 0.4 mm; and cutting width *a_e_* = 0.3 mm. 

The micro-texture was processed onto the cutter using a fiber laser. The use parameters for this were: wave length *λ* = 1064 nm; maximum output power *P* = 70 W; frequency *F* = 3.14 kHz; and scanning speed *V* = 500 m/s. An inappropriate choice of micro-texture parameters can accelerate wear on the surface of the front blade, and increase the likelihood of the cutting-edge tipping. In the case of our experiment, the range for the diameter of the micro-pits was 30–60 μm, the range for their depth was 40–70 μm, the range for their interval (or spacing) was 125–200 μm, and the range for their distance from the cutting edge was 90–120 μm [27]. After laser fabrication of micro-texture cutters, the surface burrs were removed by polishing with 2000 mesh metallographic sandpaper, and then cleaned by ultrasonic waves. The cleaning agent was acetone.

The effects of the different parameters relating to hardening and surface roughness were analyzed by using an orthogonal test with four factors and four levels (see Table 3). 

## 5. Analysis of the Test Results

### 5.1. Influence of Micro-Texture Parameters on Surface Roughness

#### 5.1.1. Test Data and Orthogonal Analysis of Surface Roughness

The titanium alloy was milled with the micro-textured ball-end milling cutter, according to the orthogonal test scheme. After changing the diameter, depth, and spacing of the micro-pit texture and the distance to the cutting edge, the surface roughness of the workpiece surface produced by the different micro-texture parameters was collected. The titanium alloy surface was processed and measured using each parameter three times, then the average surface roughness was taken as the test result. Before measuring the machined surface roughness, in order to eliminate other interference, the workpiece needs to be thoroughly cleaned with alcohol. Measurements of the machined surface roughness was performed white light interferometer. The test data and results are shown in Table 4.

The experimental results in the orthogonal table were then processed and analyzed by using extreme difference analysis, which is a simple and practical method for arriving at the relative order of influence for different parameters in a test [28]. The outcome of the extreme analysis is shown in Table 5. *fi* (*i* = 1, 2, 3, 4) represents the influence of the four factors on the diameter, depth, interval and distance from the cutting edge for the micro-pits; *Ki* (*i* = 1, 2, 3, 4) represents the sum of the experimental data for the impact of factor I on the level of *j* (*j* = 1, 2, 3, 4). The average value, *ki*, (*i* = 1, 2, 3, 4) represents the sum of the corresponding test data; R represents the extreme value for each factor. The test results were used to determine the order and change law for the influence of the various factors on the surface roughness.

The range of the four factors was obtained by using range analysis. The order of influence of the micro-pit texture parameters on the surface roughness was as follows: diameter > spacing > depth > cutting edge distance. This indicates that the diameter of the micro-pits has the greatest influence on the surface roughness, whilst the distance between the cutting edge and the micro-pits has the least. In that case, to reduce the effect of the cutter on the surface roughness, the micro-pit diameter should be taken as a priority when milling titanium alloy.

#### 5.1.2. Analysis of the Influence of Micro-Texture Parameters on Surface Roughness

When using micro-textured ball-end milling cutters on titanium alloy, the placement of a micro-texture at the tool-chip interface can reduce the actual contact area, thereby reducing the friction between the contact surface and the chip. This, in turn, helps to reduce the cutting force and cutting temperature. Plastic deformation is then reduced for the cutting area on the metal’s surface, thereby also reducing the surface roughness. As the pressure and temperature is relatively high near the cutting edge (the internal friction area), a non-woven tool becomes seriously bonded, increasing the adhesion on the edge of the chip. That being the case, for non-woven cutters, the insertion of a micro-texture on the surface effectively reduces the bonding phenomenon. Micro-textured tools become covered with adhesive material around the small holes near the cutting edge. However, there is less adhesion near the outer edge of the micro-pits. The reason for this is that, in the process of metal cutting, the placement of the micro-texture changes the chip flow direction, making adhesion on the machined surface relatively light. it can be concluded, then, that micro-textures have a significant influence on the surface roughness.

Figure 5 shows the influence of the micro-pit diameter on the surface roughness. It can be seen that the surface roughness tends to decrease as the diameter of the micro-pits is increased. When the diameter of the micro-pits is 60 µm, the surface roughness is 0.504 µm, which equates to a good surface quality. This is because, during the cutting process, the micro-pit texture captures impurities such as chips and abrasive particles. So, as the diameter increases, the ability to store these impurities is enhanced. Figure 6 is a scanning electron microscope (SEM) image for a micro-pit with a diameter of 60 µm. As the diameter increases, the angle between the edge of the pit and the chip flow varies from 0° to 90°. When the chip flows over the micro-pits, it is squeezed into them by the pressure. As the chip is pushed into the micro-pit, it is subjected to secondary “cutting” by the micro-pit’s edge. The larger the micro-pit diameter, the more obvious the secondary “cutting” phenomenon becomes.

Figure 7 shows the effect of micro-pit depth on the surface roughness. It can be seen that, as the depth of the pit increases, the surface roughness decreases. When the depth of the pit is 70 µm, the surface roughness is 0.555 µm and the surface quality is good. When the depth of the micro-pit is 40 µm, the surface roughness is 0.627 µm, and the surface quality is poor. This is because the larger cutting force and the increase in the degree of plastic deformation increases the surface roughness value. Something else to take into account is that there is a recasting phenomenon when a micro-texture is processed by a laser. This phenomenon is caused by the Gaussian distribution of the energy produced by the laser, and its attenuation in other directions, which gives each micro-pit a slight taper. When the depth of the micro-pits is too small, the tapering becomes more obvious, limiting their capacity to store impurities. In that case, it is evident that if the depth of the micro-pits is too small, the effectiveness of the cutter is diminished, and it is hard to obtain a good surface quality.

Figure 8 shows the effect of micro-pit spacing on the surface roughness. Here, we can see that the surface roughness initially increases, then decreases as the distance between the micro-pits is increased. The surface roughness is small and the surface quality good when the distance between the micro-pits is 200 µm. Figure 9 shows the surface morphology of the workpiece for different micro-pit spacings. It can be seen from the figure that the surface quality of the workpiece is better when using a micro-textured tool. Figure 9b shows the surface quality of the workpiece when the spacing is 200 µm. There are no obvious protuberances or depressions on the surface. Figure 9a shows the surface quality of the workpiece when the spacing is 175 µm. Here, it can be seen that there are obvious protrusions and dents. This is because front face bonding on the tool is more serious at this level of spacing, which causes bonding on the surface of the workpiece as well. This directly leads to an increase in surface roughness, and a deterioration of surface quality. So, to obtain good surface quality, the gap between the micro-pits should not be too small.

Figure 10 shows the influence of the distance between the micro-texture and the cutting edge on the surface roughness. In this case, we can see that the surface roughness initially increases, then decreases, then shows an upward trend. When the distance between the micro-pits and the cutting edge is 110 µm, the surface roughness is at its lowest. This is because the texture is positioned close to the cutting edge, which undermines the strength of the tool. In machining processes, if the distance between the micro-pits and the cutting edge is too small, it causes a concentration of stress. As a result, the cutting leaves a hard point on the micro-avalanche blade. This hard point plows into the machined surface during cutting. At the same time, if the micro-texture is too far away from the cutting edge, it does not play a full part in the cutting process. So, a distance between the micro-pits and the cutting edge of 110 µm, produces the best surface quality.

### 5.2. Influence of Micro-Texture Parameters on Work Hardening

#### 5.2.1. Test Data and an Orthogonal Analysis of Work Hardening

The micro-hardness of the machined surfaces was also measured using the same set of parameters. The experimental results are shown in Table 6.

The orthogonal experiment test results were again processed and analyzed using extreme difference analysis. Table 7 shows the extreme analysis results. As before, *fi* (*i* = 1, 2, 3, 4) represents the effect of the four factors and *Ki* (*i* = 1, 2, 3, 4) represents the sum of level *j* (*j* = 1, 2, 3, 4) for the *i* impact factor test data. *ki* (*i* = 1, 2, 3, 4) represents the average for the sum of the corresponding test data. *R* represents the extreme value for each factor. The order of influence for the main and secondary factors, and the law of change for work hardening was determined using these results.

The main and secondary order of influence of the micro-pit texture parameters on the surface hardening of titanium alloy surface is obtained by the extreme difference analysis as follows: micro-pit diameter > micro-texture distance from the blade > micro-pit depth > micro-pit spacing. So, the diameter of the micro-pits has the greatest influence on work hardening, whilst the distance between them has the least.

#### 5.2.2. Analysis of the Influence of Micro-Texture Parameters on Work Hardening

The placement of a micro-texture on the tool surface can reduce the contact area between the tool and the chip when milling titanium alloy. This reduces the friction at the tool-chip interface, thus also reducing the cutting force, the cutting temperature and the degree of deformation of the cutting surface. This also helps to reduce the level of surface hardening.

As the thermal conductivity of titanium alloy is very low and its heat transfer performance is poor, when a chip is detached from the workpiece surface, the surface heat generated by the cutting will cause the surface temperature to rise. However, the contact length between the chip and the cutter changes when the texture diameter is changed and this can have an effect on the degree of work hardening and the surface hardening phenomenon. As is shown in Figure 11, when the diameter is 30 μm, the work hardening is at its most serious. As the diameter increases, the work hardening phenomenon attenuates.

With an increase in the depth of the micro-pits, the work hardening phenomenon appears to decrease at first, and then it increases, before finally decreasing again. The pattern of influence is shown in Figure 12. When the depth is 40 μm, the degree of surface hardening is at its largest. As the cutting force is at its maximum here, the degree of plastic deformation is also at its maximum, which leads to an increase in the degree of hardening. So, in order to reduce the cutting force and to reduce the degree of surface hardening properly, the micro-pit depth needs to be increased.

The effect of the micro-pits spacing on the micro-hardness is shown in Figure 13. The degree of hardening is 147%, 138%, 136%, and 138%, respectively, for when the spacing of the pits is 125 μm, 150 μm, 175 μm, and 200 μm. That being so, the degree of surface hardening can be best reduced by having a distance between the micro-pits of more than 125 μm.

Looking at Figure 14, when the distance between the micro-pits and the cutting edge is increased, the degree of surface hardening shows a decreasing trend. This is because, when the micro-texture is closer to the cutting edge, the tool strength is decreased, which aggravates tool wear, increases the cutting force, extends the amount of surface plastic deformation, and, thus, the degree of work hardening.

In the early and middle stages of milling process, the influence of the micro-texture ball-end milling cutter on surface hardening is not obvious, compared with non-texture ball-end milling cutter. However, with the increase of cutting stroke, the existence of the micro-texture increases the heat dissipation area of the front face of the ball-end milling cutter, and decreases the friction force, so that the cutting temperature is lower than that of a non-texture ball-end milling cutter. The oxidation degree of the titanium alloy decreases, so the surface hardening degree decreases with the implantation of the micro-texture.

## 6. Prediction Model and the Test of its Significance

### 6.1. Surface Integrity Prediction Model

Taking the orthogonal experiment reported above as its basis, we are going to establish a regression model of work hardening that takes into account the diameter, depth, spacing and distance from the cutting edge of a micro-pit texture. The mathematical formulation of this model is as follows:(3)$$HV=Cd^{\alpha 1}h^{\alpha 2}L_{1}^{\alpha 3}L_{2}^{\alpha 4}$$

The logarithm of the two sides of the formula can be obtained at the same time.
(4)$$LgHV=LgC+\alpha_{1}Lgd++\alpha_{2}Lgh+\alpha_{3}LgL_{1}+\alpha_{3}LgL_{2}$$

Taking the order y=LgHV, $\alpha_{0}=LgC$, $x_{1}=Lgd$, $x_{2}=Lgh$, $x_{3}=LgL_{1}$, and $x_{4}=LgL_{2}$, formula (4) can be transformed into a linear equation as follows:(5)$$y=\alpha_{0}+\alpha_{1}x_{1}+\alpha_{2}x_{2}+\alpha_{3}x_{3}+\alpha_{4}x_{4}$$

Taking into consideration formula (5) and the data from the orthogonal tests, a multiple linear regression equation can be established, using the least square method:(6)$$\begin{array}{l} y_{1}=\alpha_{0}+\alpha_{1}x_{11}+\alpha_{2}x_{12}+\alpha_{3}x_{13}+\alpha_{4}x_{14}+\varepsilon_{1}\\ y_{2}=\alpha_{0}+\alpha_{1}x_{21}+\alpha_{2}x_{22}+\alpha_{3}x_{23}+\alpha_{4}x_{24}+\varepsilon_{2}\\ \vdots \\ y_{16}=\alpha_{0}+\alpha_{1}x_{161}+\alpha_{2}x_{162}+\alpha_{3}x_{163}+\alpha_{4}x_{164}+\varepsilon_{16} \end{array}$$

The general expression of the regression equation is as follows:(7)$$y_{1}=b_{0}+b_{1}x_{1}+b_{2}x_{2}+b_{3}x_{3}+b_{4}x_{4}$$

Due to the large number of test groups, the quantity of test data and the time involved in processing it, EXCELL software was used to perform a multiple linear regression analysis of the experimental data. From this, we obtained a work-hardening prediction model for micro-textured ball-end milling cutters when milling titanium alloy. It can be expressed as follows:(8)$$HV=10^{4.4943}d^{-0.2430}h^{-0.2047}L_{1}^{-0.1103}L_{2}^{-0.4606}$$

The surface roughness experimental data was processed in a similar way. The prediction model for surface roughness is:(9)$$Ra=10^{1.0318}d^{-0.3511}h^{-0.1937}L_{1}^{-0.2394}L_{2}^{0.0885}$$

### 6.2. Test of the Significance of the Prediction Model

In order to ensure the reliability of the prediction model for surface roughness and work hardening, and to check the linear relationship between the target and the parameter values, we conducted a significance test for the regression equation. This assesses whether the influence of all of the independent variables upon a dependent variable is significant. Variance analysis was carried out on all of the orthogonal test data. Table 8 and Table 9 present the results of the variance analysis.

Table 8 and Table 9 show the data from the statistical analysis. This data was used to test the significance of the prediction models. The number of experiments *n* = 16, the number of independent variables *m* = 4 and the significance level was set at 0.05. For both prediction models the significance *F* is less than 0.5: *FRa* (*m*, *n* − *m* + 1) = *F*0.95 (4, 11) = 3.68 > 3.36; *FHV* (*m*, *n* − *m* + 1) = *F*0.95 (4, 11) = 6.46 > 3.36. As a result, the linear relationship between the target values and the parameter values can be seen to hold and the surface roughness and work hardening prediction models for a micro-textured ball-end milling cutter can both be considered to be significant.

## 7. Conclusions

By engaging in a milling experiment where a micro-textured ball-end milling cutter was used for the machining of titanium alloy, we obtained a range of data relating to both the surface roughness and work hardening. The influence of various micro-pit parameters for the texture were analyzed, and the following conclusions were obtained:The order of influence for the texture parameters in relation to the surface roughness is: micro-pit diameter > micro-pit interval > micro-pit depth > micro-pit distance from the cutting edge. The primary and secondary micro-pit texture parameters influencing work hardening are: micro-pit diameter > micro-pit distance from the cutting edge > micro-pit depth > micro-pit interval.The smaller the diameter of the micro-pits, the more serious the work hardening phenomenon becomes. As the micro-pit depth is increased, the hardening phenomenon decreases, increases, and then decreases again more gradually. When the micro-pit distance from the cutting edge increases, the degree of work hardening decreases. Outside of this, when the micro-pit spacing is increased, the degree of work hardening can also be reduced.Regression analysis carried out on the orthogonal test results established prediction models for work hardening and surface roughness. The reliability of these models was confirmed by comparison between values from the experiments and the values predicted by the models.

## Figures and Tables

**Figure 1 micromachines-10-00021-f001:**
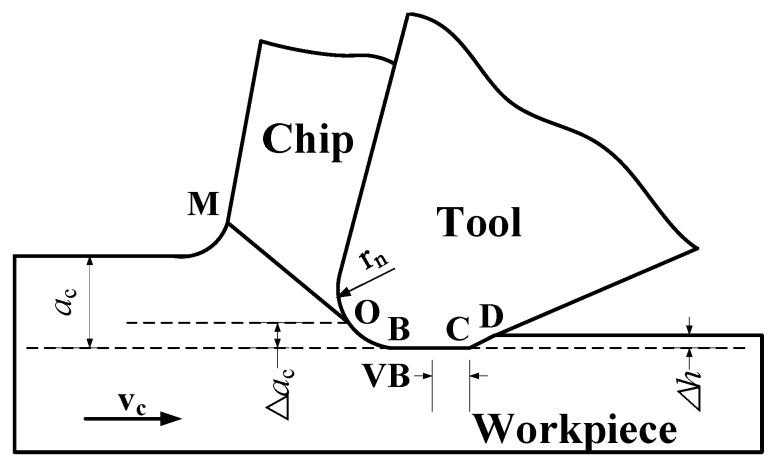
Formation of the machined surface.

**Figure 2 micromachines-10-00021-f002:**
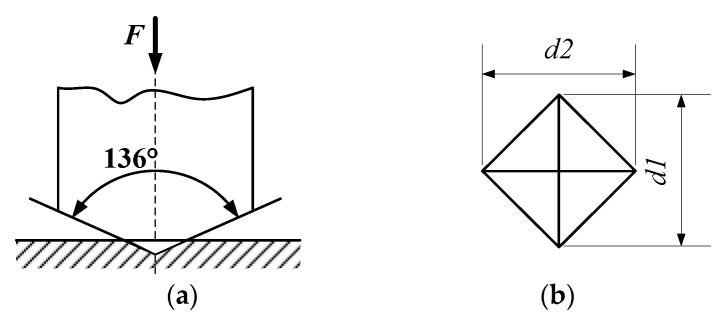
Indentation-based hardness test. (**a**) Pressure head; (**b**) vivtorinox hardness indentation.

**Figure 3 micromachines-10-00021-f003:**
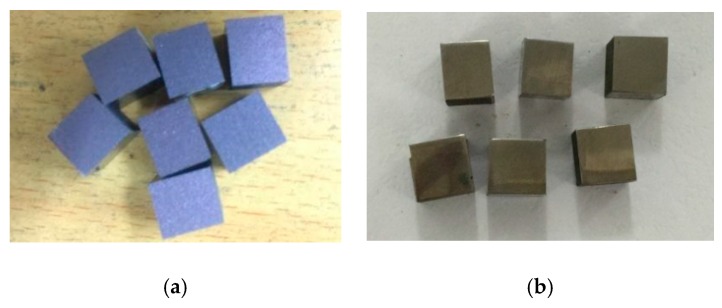
Titanium alloy samples. (**a**) Samples after wire cutting; (**b**) samples after grinding and polishing.

**Figure 4 micromachines-10-00021-f004:**
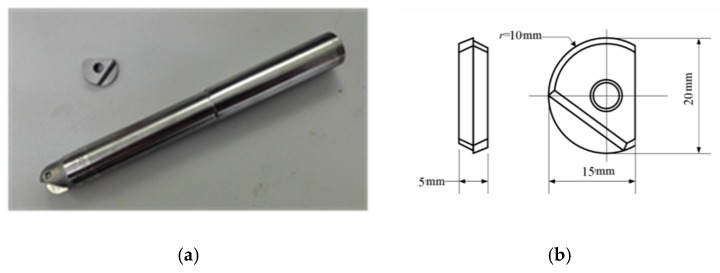
The tool shank and insert. (**a**) Physical graph; (**b**) dimensional figure.

**Figure 5 micromachines-10-00021-f005:**
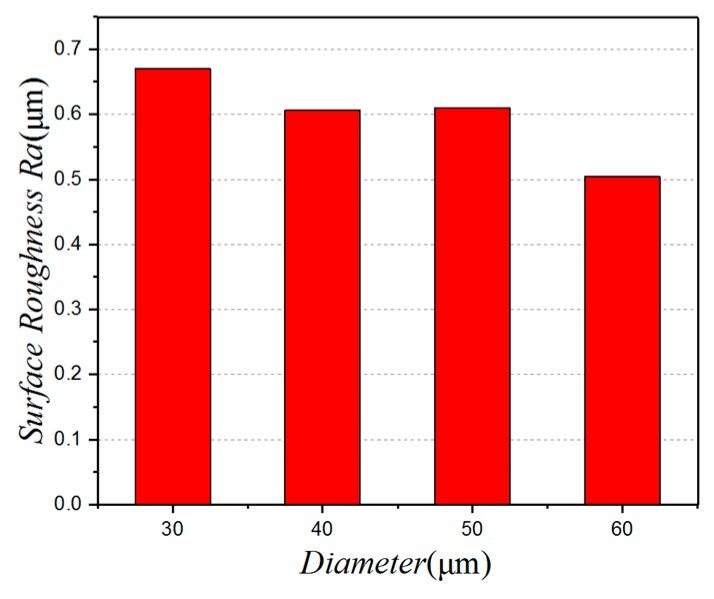
The relationship between the diameter and the surface roughness.

**Figure 6 micromachines-10-00021-f006:**
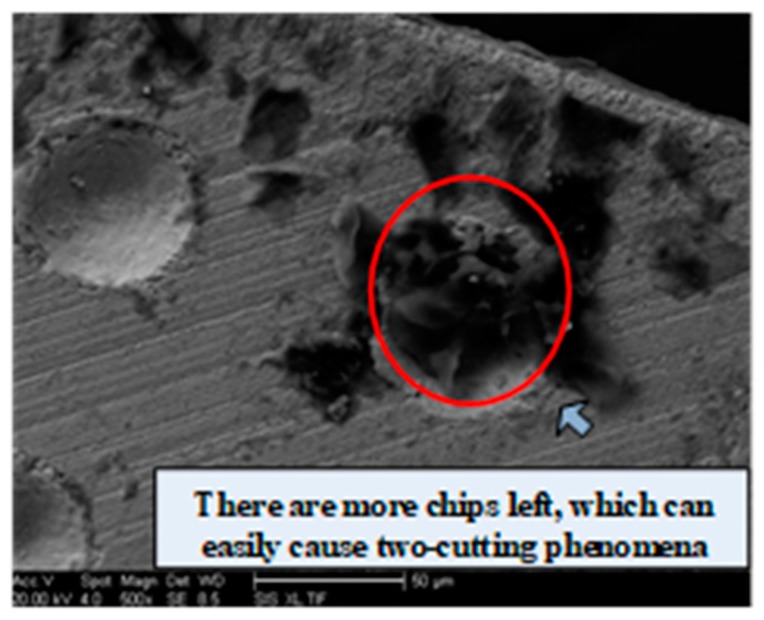
SEM image for a micro-pit with a diameter of 60 µm.

**Figure 7 micromachines-10-00021-f007:**
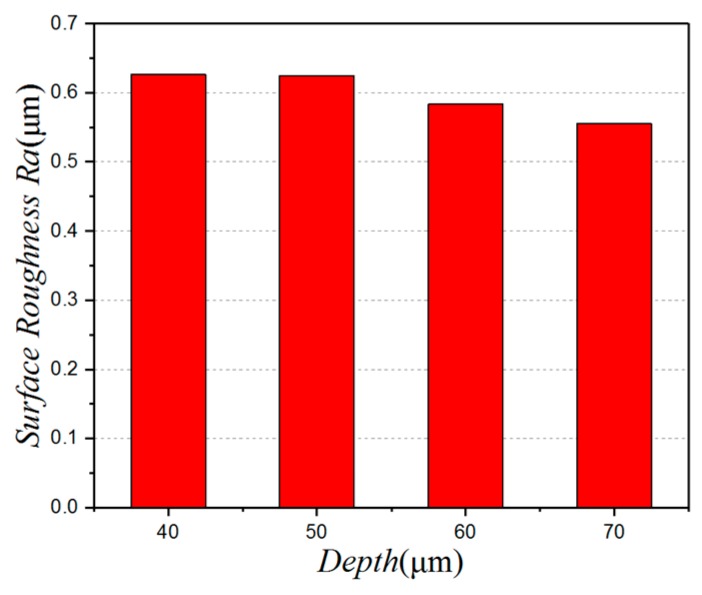
The relationship between the depth and the surface roughness.

**Figure 8 micromachines-10-00021-f008:**
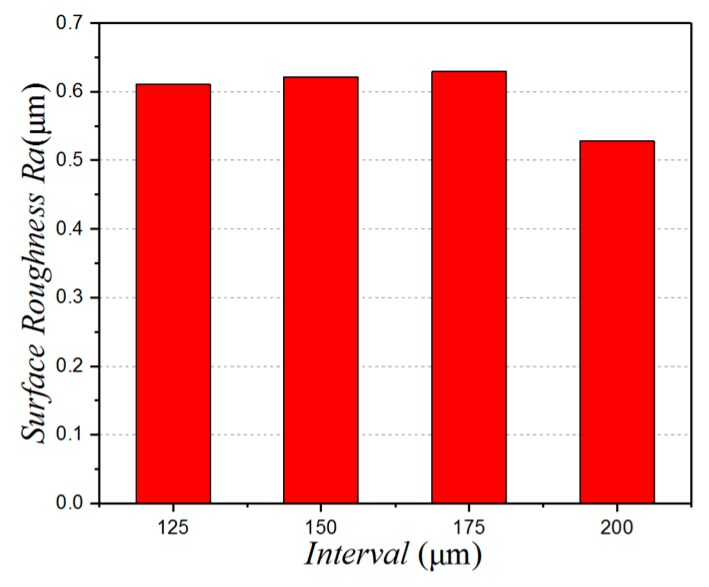
The relationship between the depth and the surface roughness.

**Figure 9 micromachines-10-00021-f009:**
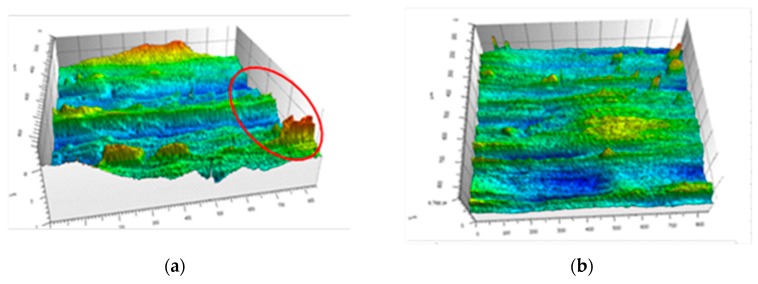
Workpiece surface morphology. (**a**) 175 µm; (**b**) 200 µm.

**Figure 10 micromachines-10-00021-f010:**
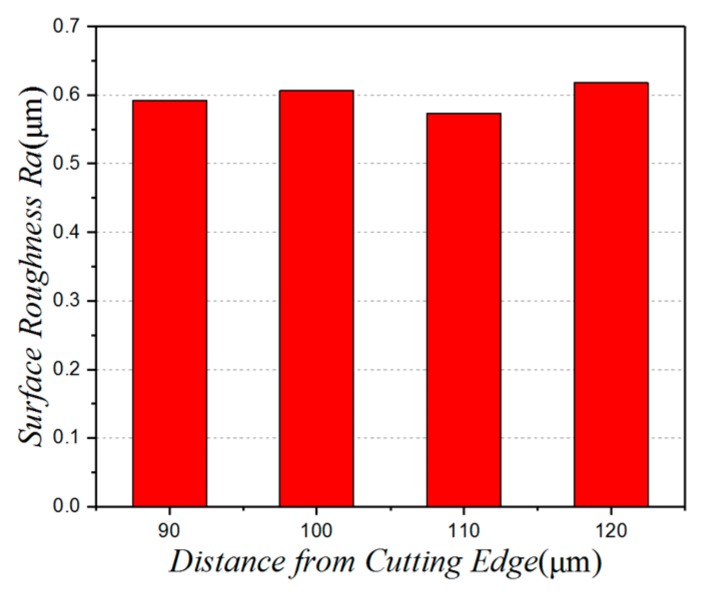
The influence of the distance between the micro-texture and the cutting edge on the surface roughness.

**Figure 11 micromachines-10-00021-f011:**
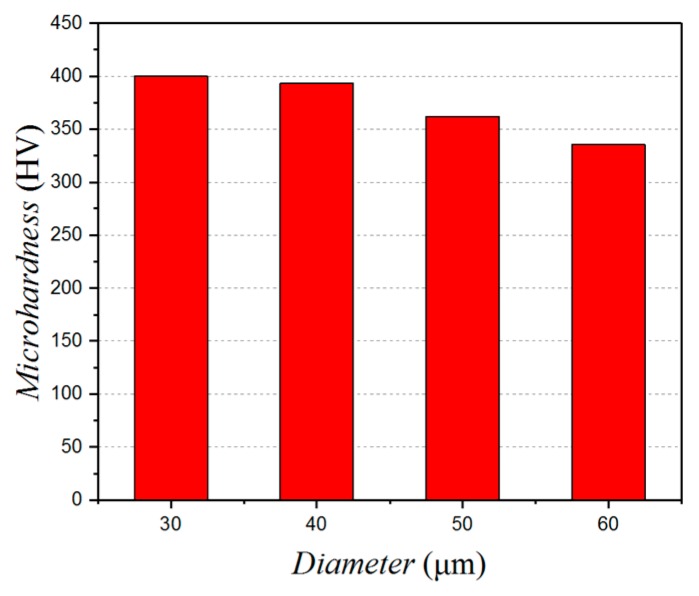
Influence of the micro-texture diameter on the surface micro-hardness.

**Figure 12 micromachines-10-00021-f012:**
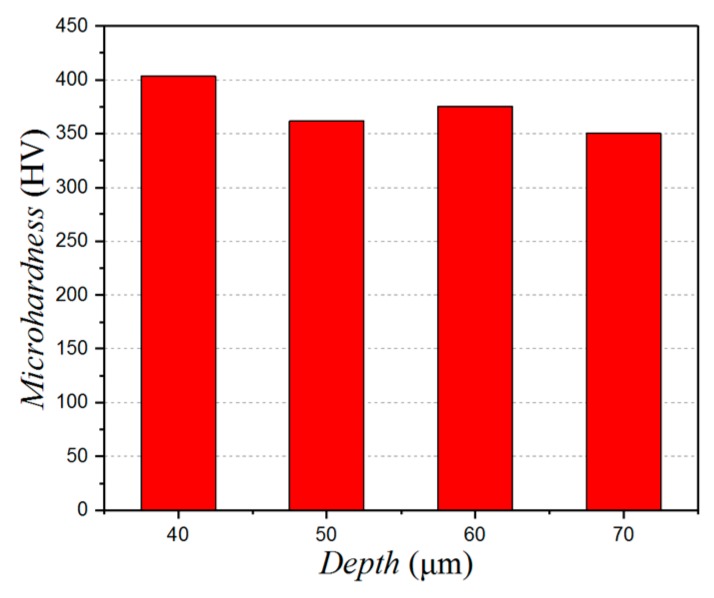
Influence of the micro-texture depth on the surface micro-hardness.

**Figure 13 micromachines-10-00021-f013:**
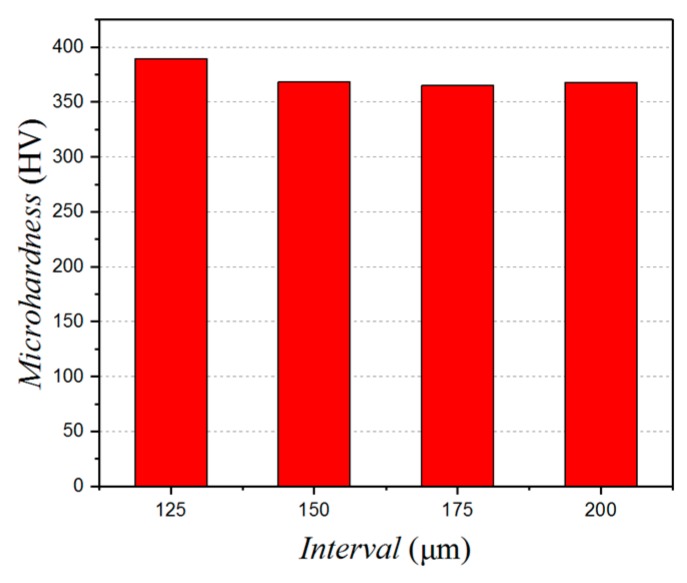
The effect of the micro-pits interval on the micro-hardness.

**Figure 14 micromachines-10-00021-f014:**
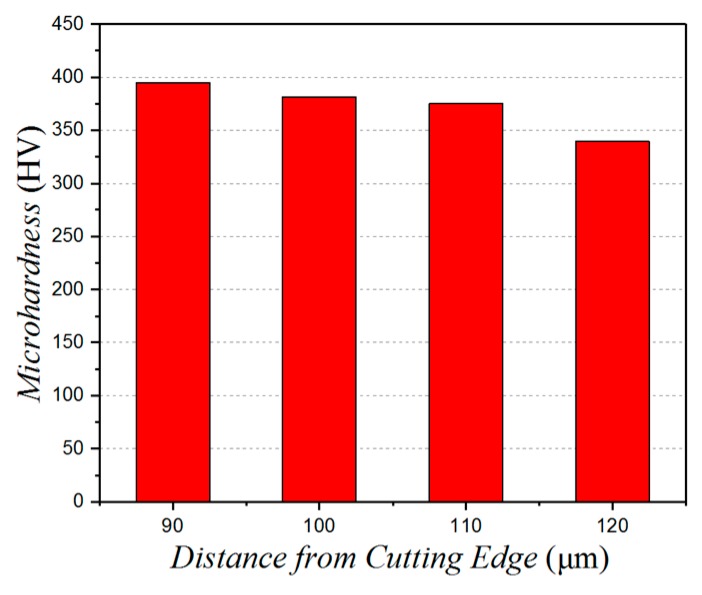
The effect of the micro-pits distance from cutting edge on the micro-hardness.

**Table 1 micromachines-10-00021-t001:** Physical and mechanical properties of Ti6Al4V.

Hardness HRC	Density kg/m^3^	Melting Point °C	Specific Heat J/(kg∙K)	Heat Transfer Coefficient W/(m·K)	Poisson Ratio v	Yield Strength MPa	Modulus of Elasticity GPa
36	4428	1605	1012	7.955	0.41	825	110

**Table 2 micromachines-10-00021-t002:** Chemical composition of Ti6Al4V (%).

Element	Al	V	Fe	C	N	H	O	Ti
Content	5.5–6.8	3.5–4.5	≤0.30	≤0.10	≤0.05	≤0.015	≤0.20	allowance

**Table 3 micromachines-10-00021-t003:** Factors and levels.

	Factor	Diameter (*D*) μm	Depth (*H*) μm	Interval (*L1*) μm	Distance from Cutting Edge (*L2*) μm
Level					
1		30	40	125	90
2		40	50	150	100
3		50	60	175	110
4		60	70	200	120

**Table 4 micromachines-10-00021-t004:** Surface roughness test data and test results.

Test Number	Diameter μm	Depth μm	Interval μm	Distance from Cutting Edge μm	Average Surface Roughness μm
1	30	40	125	90	0.7339
2	30	50	150	100	0.77
3	30	60	175	110	0.6373
4	30	70	200	120	0.5397
5	40	40	150	110	0.5955
6	40	50	125	120	0.6406
7	40	60	200	90	0.5569
8	40	70	175	100	0.6306
9	50	40	175	120	0.7305
10	50	50	200	110	0.5695
11	50	60	125	100	0.579
12	50	70	150	90	0.56
13	60	40	200	100	0.4461
14	60	50	175	90	0.5186
15	60	60	150	120	0.5606
16	60	70	125	110	0.4906

**Table 5 micromachines-10-00021-t005:** Results of the extreme difference analysis.

Test Parameters	*f1*	*f2*	*f3*	*f4*
*K1*	2.6809	2.506	2.4441	2.3694
*K2*	2.4263	2.4987	2.4861	2.4257
*K3*	2.439	2.3338	2.517	2.2929
*K4*	2.0159	2.2209	2.1122	2.4714
*k1*	0.670225	0.6265	0.611025	0.59235
*k2*	0.6059	0.624675	0.621525	0.606425
*k3*	0.60975	0.58345	0.62925	0.573225
*k4*	0.503975	0.555225	0.52805	0.61785
*R*	0.16625	0.071275	0.1012	0.044625

**Table 6 micromachines-10-00021-t006:** Surface-hardness test data and test results.

Test Number	Diameter μm	Depth μm	Interval μm	Distance from Cutting Edge μm	Surface Micro-Hardness HV
1	30	40	125	90	484.8
2	30	50	150	100	392.11
3	30	60	175	110	406.92
4	30	70	200	120	316.13
5	40	40	150	110	397.73
6	40	50	125	120	375.32
7	40	60	200	90	411.91
8	40	70	175	100	386.66
9	50	40	175	120	349.88
10	50	50	200	110	363.41
11	50	60	125	100	367.26
12	50	70	150	90	367.17
13	60	40	200	100	380.13
14	60	50	175	90	316.13
15	60	60	150	120	315.35
16	60	70	125	110	330.93

**Table 7 micromachines-10-00021-t007:** Results of the surface hardening extreme analysis.

Test Parameters	*f1*	*f2*	*f3*	*f4*
*K1*	1599.96	1612.54	1558.31	1580.06
*K2*	1571.62	1446.97	1472.36	1526.16
*K3*	1447.72	1501.44	1459.59	1498.99
*K4*	1342.54	1400.89	1471.58	1356.68
*k1*	399.99	403.135	389.5775	395.015
*k2*	392.905	361.7425	368.09	381.54
*k3*	361.93	375.36	364.8975	374.7475
*k4*	335.635	350.2225	367.895	339.17
*R*	64.355	52.9125	24.68	55.845

**Table 8 micromachines-10-00021-t008:** Variance analysis of surface roughness (Ra).

Source of Variance	SS	DF	MS	F	Significance F
Ra	0.0354	4	0.0089	3.68	0.0388
Residual	0.0265	11	0.0024	—	—
Total	0.0619	15	—	—	—

**Table 9 micromachines-10-00021-t009:** Variance analysis of surface hardness (HV).

Source of Variance	SS	DF	MS	F	Significance F
HV	0.025944	4	0.006486	60458596	0.006304926
Residual	0.011046	11	0.001004		
Total	0.03699	15

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
