# Peer review of "The Surface Integrity of Titanium Alloy When Using Micro-Textured Ball-End Milling Cutters"

_micromachines, 2018, doi:10.3390/mi10010021_

Round 1
Reviewer 1 Report
Micromachines-402323-peer-review-v1:
The surface integrity of titanium alloy when using micro-textured ball-end milling cutters
Reviewers Comments
General comments:
This research article considers the effects of textured surfaces when applied to cutting tools and the effect on roughness and work hardening of changing the surface texture geometry.
Some re-writing is required throughout to improve the standard of English for publication.
More consideration needs to be made on how the surface roughness is measured, machinery, technique, filters, cut offs etc. This will play a large part in the results gained and is required in order for your results to be valid.
It is also necessary to provide a control for the surface roughness analysis (an unfeatured tool) as this would form the benchmark for the contribution of the textured surface.
To further increase the validity of the work presented here error bars are required throughout. Results are not valid based on one sample measurement alone and the spread of the data is required to understand the actual differences perceived.
The reviewer wonders as no post finishing of the tooling was shown to be carried out post the texturing procedure, is it worth considering, as it is well known that laser dimpling causes burs above the surface (visible in figure 7), that this could inhibit or contribute to some of the findings here ?
The concept of the paper is good and is currently relevant to industry. However scientific integrity needs to be improved throughout.
Specific Comments:
Line 32 – Please change the term “and so on” this is not scientific wording. Please be concise about the benefits and use references to help expand the point.
Line 36 – Please rephrase. I don’t think these factors limit the development of titanium alloy technology. There are many other manufacturing technologies that have no issue in machining titanium. It is limiting use of traditional manufacturing methods in working Ti.
Line 39 – Please add references to the benefits found in prior art.
Line 44 – Again add references where alternative technology has been used.
Line 65 to 67 – Please rephrase. This is rather confusing to the reader. Do you mean the micro hardness indentation size is this ? Hardness isn’t usually measured in microns, this suggests a geometric quantity.
Line 86 – if rn is a notation please place in brackets, this also applies for other notation throughout when it is described for the first time without being the point of the sentence.
Lines 108 to 114 – Please rephrase or remove this paragraph. The reader will be aware of what cutting is. There is a valid point here but it can be incorporated into the paragraph below as it currently repeats the point.
Line 124 – Should it not make any difference to the surface integrity of the parts ?
Line 131 – See earlier comment around using the term “and so on” and remove throughout
Equation 2 – Is this a known formula ? If so reference it. Also where do the two constant values come from ? Again references will suffice unless simple to explain.
Figure 4 – Please add unit of measurement
Table 4 – There is no unit for surface roughness and what type of surface roughness this is.
Figure 6 – No error bars are present on this graph. Was only one measurement taken ?
Figure 10 – The quality of these images must be improved. It is difficult for the reader to take anything away from these images.
Author Response
Dear reviewers:
We appreciate the thorough reviews and positive comments provided by the reviewers on our submitted paper. According to these objective and professional suggestions, careful revisions have been made. Detailed replies to your comments are presented as follows.
Modification description
Reviewer:
This research article considers the effects of textured surfaces when applied to cutting tools and the effect on roughness and work hardening of changing the surface texture geometry.
Some re-writing is required throughout to improve the standard of English for publication.
More consideration needs to be made on how the surface roughness is measured, machinery, technique, filters, cut offs etc. This will play a large part in the results gained and is required in order for your results to be valid.
It is also necessary to provide a control for the surface roughness analysis (an unfeatured tool) as this would form the benchmark for the contribution of the textured surface.
To further increase the validity of the work presented here error bars are required throughout. Results are not valid based on one sample measurement alone and the spread of the data is required to understand the actual differences perceived.
The reviewer wonders as no post finishing of the tooling was shown to be carried out post the texturing procedure, is it worth considering, as it is well known that laser dimpling causes burs above the surface (visible in figure 7), that this could inhibit or contribute to some of the findings here ?
The concept of the paper is good and is currently relevant to industry. However scientific integrity needs to be improved throughout.
Answer: The authors have double checked the paper, and the errors have been corrected. The wordy parts have been revised, and a lot of unsuitable logic descriptions were also improved. The surface roughness data are measured by white light interferometer three times, and the average value of three times is taken as the experimental data. Before the measurement, the processed surface was wiped with alcohol. The material in Figure 7 is chips generated during milling. Surface burrs have been removed after treatment, and I've added these changes to the article.
1) Line 32 – Please change the term “and so on” this is not scientific wording. Please be concise about the benefits and use references to help expand the point.
Answer: I've made changes.
2) Please rephrase. I don’t think these factors limit the development of titanium alloy technology. There are many other manufacturing technologies that have no issue in machining titanium. It is limiting use of traditional manufacturing methods in working Ti.
Answer: I have revised my statement.
3) Please add references to the benefits found in prior art.
Answer: Reference has been added.
4) Again add references where alternative technology has been used.
Answer: Reference has been added
5) Line 65 to 67 – Please rephrase. This is rather confusing to the reader. Do you mean the micro hardness indentation size is this ? Hardness isn’t usually measured in microns, this suggests a geometric quantity.
Answer: 100 micron refers to the depth of hardened case. It Has been modified to complete.
6) Line 86 – if rn is a notation please place in brackets, this also applies for other notation throughout when it is described for the first time without being the point of the sentence.
Answer: Rn has placed in brackets
7) Lines 108 to 114 – Please rephrase or remove this paragraph. The reader will be aware of what cutting is. There is a valid point here but it can be incorporated into the paragraph below as it currently repeats the point.
Answer: This paragraph has been deleted
8) Line 124 – Should it not make any difference to the surface integrity of the parts ?
Answer: whether the choice of cutting amount is reasonable makes a great deal of difference to the surface integrity of the parts.
9) Line 131 – See earlier comment around using the term “and so on” and remove throughout
Answer: “and so on” has been deleted.
10) Equation 2 – Is this a known formula ? If so reference it. Also where do the two constant values come from ? Again references will suffice unless simple to explain.
Answer: References to the formula have been added.
11) Figure 4 – Please add unit of measurement
Answer: Units have been added to the figure.
12) Table 4 – There is no unit for surface roughness and what type of surface roughness this is.
Answer: Roughness units have been added to the table.
13) Figure 6 – No error bars are present on this graph. Was only one measurement taken ?
Answer: The experiment was measured three times. The data obtained are the average of three measurements. There are corresponding explanations in the article.
14) Figure 10 – The quality of these images must be improved. It is difficult for the reader to take anything away from these images.
Answer: I've already processed the pictures.
Thank you again for the comments from all the reviewers. If there are any more shortcomings, please inform us.
Best wishes
Reviewer 2 Report
Do absolutely all changes compulsory in attached file

Author Response
Dear reviewer:
We appreciate the thorough reviews and positive comments provided by you on our submitted paper. According to these objective and professional suggestions, careful revisions have been made. Thank you very much for your comments on my research. I will seriously refer to the journals and articles you recommended for me in future research.
Thank you again for the comments. If there are any more shortcomings, please inform us.
Best wishes

Reviewer 3 Report
(1) The manuscript presents experimental studies on the effect of micro textures on the surface integrity. The fundamental mechanism of how the surface micro texture changes the surface hardness should be described based on the experimental data.
(2) How the micro texture influences the surface microstructure property?
Author Response
Dear reviewers:
We appreciate the thorough reviews and positive comments provided by you on our submitted paper. According to these objective and professional suggestions, careful revisions have been made. Detailed replies to your comments are presented as follows.
Modification description
Reviewer:
1) The manuscript presents experimental studies on the effect of micro textures on the surface integrity. The fundamental mechanism of how the surface micro texture changes the surface hardness should be described based on the experimental data.
Answer: In the early and middle stages of milling process, the influence of micro-texture ball-end milling cutter on surface hardening is not obvious compared with non-texture ball-end milling cutter. However, with the increase of cutting stroke, the existence of micro-texture increases the heat dissipation area of the front face of ball-end milling cutter and decreases friction force, so cutting temperature is lower than that of non-texture ball-end milling cutter. The oxidation degree of titanium alloy decreases, so the surface hardening degree decreases with the implantation of micro-texture.
2) How the micro texture influences the surface microstructure property?
Answer: Micro-texture can store fine chips and abrasives and other impurities, thus improving the surface quality of the workpiece, and the micro-texture reduces the contact area of the chips, thereby reducing wear and tear.
Thank you again for the comments from both yourself and other reviewers. If there are any more shortcomings, please inform us.
Best wishes
Round 2
Reviewer 1 Report
The reviewer believes there is still some issues of scientific integrity with only reporting the effects of a single experiment and not considering an appropriate sample size. This should be something the authors consider in there future work.
However in this case the reviewers comments, for the most part, have been addressed with only a few minor text issues still apparent.
Author Response
We appreciate the thorough reviews and positive comments provided by both yourself and the reviewers on our submitted paper. According to these objective and professional suggestions, careful revisions have been made.
Q:The reviewer believes there is still some issues of scientific integrity with only reporting the effects of a single experiment and not considering an appropriate sample size. This should be something the authors consider in there future work.
However in this case the reviewers comments, for the most part, have been addressed with only a few minor text issues still apparent.
A:In the previous research of our group, the shape and size parameters of micro-texture were optimized. In this paper, the surface quality of the workpiece in milling titanium alloy with micro-texture ball-end milling cutter was studied under the optimized micro-texture form and size parameters. I have added references to the article.
We will conduct in-depth research on the content of this article in the follow-up work, and revise the existing problems in this article again. Thank you very much!
Reviewer 2 Report
Paper must be improved now, not in the next reserach.
Please, do and resubmit again
Author Response
We appreciate the thorough reviews and positive comments provided by both yourself and the reviewers on our submitted paper. According to these objective and professional suggestions, careful revisions have been made.
Q: Other serious drawback: people from microscales always forget good works in meso or even conventional scales, in which there are not one but many models to be migrated to micro milling. See for instance works by Uriarte et al., some of them veru key in micromilling: stiffness values for microtools, error budgets, and other. Your work is mainly experimental, but at least the scope of state of the art must be complete.
A:First of all, I must explain that micro-milling refers to the processing of micro parts, using small cutting depth and small feed to process parts. In this paper, micro texture is inserted into the front face of a medium-scale tool to improve the performance of the tool, so that the surface quality of the workpiece is better than that of a common tool. Because the insertion of micro structure improves the overall situation of cutting tools and workpieces, it can be said that micro structure plays a good role in cutting.
Q: The surface roughness of a part or product strongly influences its properties and functions including abrasion and corrosion resistance, tribological properties, and optical properties as well as the visual impression the customer desires. Therefore, in industrial manufacturing grinding and polishing techniques are widely used to reduce the roughness of surfaces. In 2017 work like this must be near to fulfil all aspects, because it is an old matter. For instance, in J. P. Huissoon, F. Ismail, A. Jafari, S. Bedi, “Automated polishing of die steel surfaces,” Advanced Manufacturing Technology, 2002, Band 19, S. 285-290 se sigth of it is developed.
A:After micro-texture is processed on the rake face of ball end milling cutter, the rake face of the cutter must be polished. The main purpose is to remove the burrs on the rake face of the tool after laser processing. In order to study the influence of micro-texture on the surface quality of titanium alloy workpiece, and then establish the prediction model of surface roughness and work hardening, it is impossible to polish the surface of the workpiece after processing, which will not be able to judge the influence of micro-texture on the surface quality and work hardening of the workpiece, and can not accurately establish the model of surface quality and work hardening of the workpiece under micro texture.
Q:By the way, 20mm diameter is mesoscale, so the below comments are full mandatory. Chenck many worjks by Salgado, Lamikiz, Lazoglu, and those below. You missed hald the state of the art.
A:The tool diameter is 20mm, but the parameters of micro texture are micron-scale, which refers to the parameters of micro texture in this paper.
Q: In the JIMTOF 2018 (now was celebrated!) there were serious concern in machining about roughness. In some cases in machine there were application of previous papers. There is one journal, the CIRP, in which several woks related with yours were proposed. I think Prof Arizmendi ones were the best (one is https://doi.org/10.1016/j.cirp.2008.03.045). The same is in The Int J Adv manufacturing, for instance the work by Olvera The International Journal of Advanced Manufacturing Technology 90 (1-4), 741-752, in which a method based on different ground is also proposed. Your state of the art is very bad. See another, Materials and Manufacturing Processes 26 (8), 997-1003, and even International Journal of Production Research 40 (12), 2789-2809, in both roughness was improved a lot. Prof Bustillo also worked with ANN in roughness in ball end milled parts.
A:The main research content of this paper is to establish a prediction model of surface integrity of titanium alloy workpiece milling with micro-texture ball-end milling cutter based on the mechanism of anti-wear and anti-wear, which can provide reference and basis for reasonable selection of micro-texture parameters. The purpose of this paper is not to optimize the surface roughness of workpiece. The reviewer gave the reference article. After careful reading, I found that the main research object of the reference article is the complex three-dimensional surface, and after dividing it into regions, the surface quality of the workpiece is predicted according to the residual height of the surface. The content of this paper is plane, and the residual and surface roughness do not belong to the same concept. This method provides a new way for the prediction of the residual height.References provided by reviewers include the measurement of surface roughness in turning, as well as changes in the composition of titanium alloys due to the insertion of micro-textures in titanium alloys. All of these provide a new direction for the follow-up research. Therefore, I have added the references used to the introduction. Thank you very much for your guidance.
If you need to revise the article again, please do not hesitate to contact me.
Reviewer 3 Report
The revised manuscript addresses the reviewers' comments and questions.
Author Response
Thank you very much ! I will continue to make further revisions to the paper.
Round 3
Reviewer 2 Report
Paper is ok